# A contemporary tool for assessing instrumental activities of daily living: Validation of a caregiver-reported scale for non-institutionalized older adults

Zainab Barakat[1,2]*, Hala Sacre[3], Sarah Khatib[3], Aline Hajj[3,4,5], Carmela BouMalham[3], Chadia Haddad[3], Rony M. Zeenny[3,6], Marwan Akel[3], Linda Abou Abbas[1,3], Marc Barakat[7], Samar Rachidi[2‡], Pascale Salameh[3,8,9,10‡]

1 Faculty of Medical Sciences, Lebanese University, Hadat, Lebanon, 2 Clinical and Epidemiological Research Laboratory, Doctoral School of Science and Technology, Lebanese University, Hadat, Lebanon, 3 Institut National de Santé Publique d'Épidémiologie Clinique et de Toxicologie-Liban (INSPECT-LB), Beirut, Lebanon, 4 Faculté de Pharmacie, Université Laval, Québec City, Québec, Canada, 5 Oncology Division, CHU de Québec-Université Laval Research Center, Québec City, Québec, Canada, 6 Department of Pharmacy, American University of Beirut Medical Center, Beirut, Lebanon, 7 Department of Psychiatry, American University of Beirut Medical Center, Beirut, Lebanon, 8 Faculty of Pharmacy, Lebanese University, Hadat, Lebanon, 9 Gilbert and Rose-Marie Chagoury School of Medicine, Lebanese American University, Beirut, Lebanon, 10 Department of Primary Care and Population Health, University of Nicosia Medical School, Nicosia, Cyprus

‡ These authors share senior authorship to this work.
* zainabbrt@outlook.com

## Abstract

### Background

Instrumental activities of daily living (IADL) refer to activities necessary for independent living, emphasizing community-related tasks. The literature has limited measurement tools that address autonomous living in contemporary communities. Consequently, our study aimed to develop, cross-culturally adapt, and evaluate the psychometric properties of a recently updated IADL scale called the Autonomy in Daily Functioning-Contemporary Scale (ADF-CS). Additionally, it sought to examine the level of agreement between informant reports and self-reports on the ADF-CS.

### Method

Following translation and cross-cultural adaptation, a first cross-sectional study was carried out among 544 family caregivers of community-dwelling older adults to assess the psychometric properties of the ADF-CS. The internal consistency of the scale was evaluated via Cronbach's alpha. Content and convergent validity, factorial analysis—including confirmatory factor analysis (CFA) and exploratory factor analysis (EFA)—and known group validity were also assessed. A second cross-sectional study involving 44 paired caregivers and care recipients was conducted to examine

**Data availability statement:** Uploaded as Supporting Information.

**Funding:** The author(s) received no specific funding for this work.

**Competing interests:** The authors have declared that no competing interests exist.

**Abbreviations:** AIADL, Advanced Instrumental Activities of Daily Living Scale; BADL, Basic Activities of Daily Living; CFA, Confirmatory Factor Analysis; CFS, Clinical Frailty Scale; EFA, Exploratory Factor Analysis; IADL, Instrumental Ictivities of Daily Living; ADF-CS, Autonomy in Daily Functioning-Contemporary Scale; ICC, Intraclass Correlation Coefficient; IQR, Interquartile Range; KMO, Kaiser-Meyer-Olkin.

the level of agreement in responses to the ADF-CS scale between caregivers and older adults. Response agreement was evaluated through intraclass and Cohen's kappa correlation coefficients.

## Results

The internal consistency of the ADF-CS and its factors was high (Cronbach's alpha between 0.83 and 0.90). The robust positive correlation between the total ADF-CS score and the ADL score supported the convergent validity of the ADF-CS Arabic version. Moreover, the statistically significant variations in ADF-CS mean scores among various age groups and some chronic disease groups supported the scale's known group validity. The EFA of the ADF-CS yielded a two-factor solution with an eigenvalue exceeding 1, explaining 63.13% of the variance. The CFA demonstrated that all the items in each component fit well with their intended constructs. Additionally, the intraclass and kappa correlation coefficient results were excellent, indicating robust agreement in the responses of caregivers and their respective older adults.

## Conclusion

The Arabic version of the ADF-CS is a reliable and valid informant-reported measure for assessing IADL in older adults living in a contemporary community.

## Introduction

The global population is experiencing a swift aging trend, with projections indicating that, by 2050, the population of individuals aged 65 and above will have more than doubled, reaching 1.5 billion and constituting 16% of the world's total population [1]. This aging trend is associated with a decline in functional ability, often exacerbated by factors such as acute illness, advanced chronic conditions, and extended hospitalization [2–4]. As functional ability declines,older adults frequently encounter challenges in performing activities of daily living (ADL), which are key life tasks essential for independent living [5,6]. ADLs are typically categorized into basic activities of daily living (BADL), which include necessary self-care tasks such as bathing, dressing, and feeding [7], and instrumental activities of daily living (IADL), which involve more complex community-related tasks such as shopping, cooking, commuting, and managing household affairs [8].

While BADLs do not necessitate high levels of cognitive attention, IADLs are more intricate and require higher cognitive functions such as memory, attention, and executive tasks, alongside physical performance [9,10]. In patients with cognitive or physical impairments, the decline in IADL usually precedes the loss of BADL capabilities, making early assessment of IADL critical for detecting functional decline and planning appropriate interventions [11]. Understanding IADL performance is vital for tailoring rehabilitation plans, organizing specialized home services, and allowing clinicians to track an individual's baseline functional status [12,13].

Despite the availability of various questionnaires for evaluating IADL, no universally accepted standard exists [14]. Among these instruments, the Lawton Instrumental Activities of Daily Living Scale (Lawton IADL), established in 1969, remains one of the most commonly adopted instruments to assess complex daily living abilities necessary for independent living in the community [4,15]. This scale is typically administered via self-reported or informant-reported (i.e., family members) questionnaires [16], which are quick and easy to administer and require minimal resources or training [17]. Informant reports on older adults' functional and cognitive abilities are particularly valuable in neurocognitive assessments [18]. In dementia evaluations, informant reports provide critical insights into early functional decline and subtle neural changes. They are also essential for assessing function and cognition in the advanced stages of the disease, where an individual's self-awareness may be significantly compromised [18].

Despite its widespread use, the Lawton IADL has significant limitations, particularly when applied to non-English-speaking and non-Western populations. While the scale was developed in English (USA) [15] and has been adapted and validated in various languages, including Spanish [19], Turkish [20], Greek [21,22], Hong Kong Chinese [23], Korean [24–26], Persian [27,28], Sinhala [29], and Malay [30], there is currently no validated Arabic version, even though the scale has been used in numerous studies [31–34]. The rapid growth of the aging population in the Arab world, coupled with the absence of a culturally adapted tool for assessing IADL, raises significant concerns by limiting accurate evaluations in Arabic-speaking populations where caregiving needs and age-related challenges differ from those in Western settings.

Furthermore, the Lawton IADL does not account for the increasing role of technology in daily life [35], despite technological skills becoming essential for independent living in modern communities, including the use of internet-connected devices like smartphones, tablets, and computers [36]. The ability to use these technologies is indispensable for older adults' daily lives, fostering independence and improving quality of life [37,38]. It also plays a key role in the development of telemedicine for dementia care [39,40].

To address this gap, the Advanced Instrumental Activities of Daily Living Scale (AIADL) was recently developed in China, building on the original Lawton scale to better reflect the evolving needs of modern life, including the use of technologies like smartphones and stored value smart cards for making electronic payments [41]. However, the latter component may not be relevant for assessing IADL in some developing countries, such as Lebanon, where smart card usage for transactions is not widespread. Moreover, updating the Lawton IADL scale to reflect modern living may require incorporating essential components that address the reliance on various technological aspects not currently covered in the AIADL scale. One such component is television viewing, as studies indicate that older adults spend more time watching TV than younger and middle-aged adults do [42]. These studies also showed that age-related increases in TV viewing exceeded those observed in other recreational activities [43,44]. Therefore, it is important to have a tool that can assess functions related to diverse technological facets in today's modern society.

Considering the limitations of current IADL assessment tools and the absence of a culturally relevant version for Arabic-speaking populations, a new tool that effectively addresses the needs of older adults in Lebanon is urgently needed. Therefore, this study addresses these gaps by developing, cross-culturally adapting, and evaluating the psychometric properties of an informant-based Arabic version of a recently updated IADL scale. On the basis of the well-known Lawton IADL scale, this new tool, the Autonomy in Daily Functioning-Contemporary Scale (ADF-CS), was designed for older adults living in community settings. The ADF-CS expands upon the Lawton IADL by incorporating additional items that reflect the technological aspects of daily living tailored for older adults in Lebanon. The present study also evaluated the agreement between informant-based and self-reported measures on the ADF-CS.

## Research design and methods

The study's methodology consisted of four phases. The first phase involved developing ADF-CS, whereas the second entailed translating and cross-culturally adapting the developed scale. The third phase evaluated the psychometric properties of the ADF-CS, including reliability, convergent validity, factorial analysis, and known group validity. Finally, the fourth

phase compared the self-reported responses on the ADF-CS with those reported by informants. The flowchart of the study phases (S1 File) illustrates the details of the entire process.

### Ethical consideration

The research protocol received approval from the Research Ethics Committee of the National Institute of Public Health, Clinical Epidemiology, and Toxicology-Lebanon (INSPECT-LB) under reference numbers 2024REC-001-INSPECT-01-13 and 2024REC-003-INSPECT-03-10 for phase III and phase IV, respectively. The study investigators adhered to the research ethics protocols delineated in the Declaration of Helsinki by the World Medical Association Assembly [45]. For phase III of the study, all caregivers provided online informed consent to confirm their voluntary participation. However, for phase IV, caregivers and care recipients provided written informed consent. In addition, for both phases, all participants had the right to decline participation, and their anonymity and confidentiality were guaranteed and respected throughout the study.

### Phase I: Development of the ADF-CS

After a thorough literature review [4,12,15,35,41,46–48], an expert panel proposed the items for the ADF-CS. The panel is composed of nine experts specializing in the following disciplines and coauthors of this work: geriatrics (n = 3), public health and epidemiology (n = 6), and scale validation (n = 4).

The panel subsequently evaluated the items' content and cultural relevance through multiple rounds of discussions via the Delphi technique until a 90% consensus was reached for each item. The discussions ultimately led to the incorporation of some items from the original Lawton IADL scale, excluding the laundry-related item, which was substituted with a new item suggested by the panel: operating household appliances such as washing machines, dryers, microwaves, and dishwashers. Additionally, new items assessing functions in contemporary community settings were introduced to the scale. Finally, the expert evaluation identified 12 items within the ADF-CS: seven derived from the original Lawton IADL and five new items that address the technological aspects of daily functioning for elderly individuals within contemporary communities.

The ADF-CS items cover a range of abilities, including the use of landline and mobile phones (ADF-CS1 and ADF-CS2, respectively), going to places beyond walking distance (ADF-CS3), shopping (ADF-CS4), housekeeping (ADF-CS5), managing medication (ADF-CS6), utilizing electronic devices such as computers and tablets (ADF-CS7), preparing foods (ADF-CS8), traveling abroad alone (ADF-CS9), using television (ADF-CS10), handling finances (ADF-CS11), and operating household appliances such as washing machines, dryers, microwaves, and dishwashers (ADF-CS12). Each item was rated on a scale of 1 (dependent), 2 (partially dependent), or 3 (independent), with the total score calculated by summing the responses across all the items. A higher score indicates a greater level of functional independence in performing IADL (S2 File).

### Phase II: Translation and cross-cultural adaptation

The ADF-CS questionnaire items were initially formulated by the expert panel in English and subsequently translated into Arabic. Additionally, since there was no Arabic culturally adapted version of the clinical frailty scale (CFS), it was translated and culturally adapted. Each scale underwent a separate translation and cross-cultural adaptation via the five-step methodology outlined in the literature [49]. More details on this process are available in S3 File. Permission was granted from Dalhousie University to use and translate the CFS into Arabic.

### Phase III: Psychometric evaluation of the ADF-CS

**Study design and participants.** A quantitative cross-sectional study targeting caregivers of older adults was carried out in Lebanon between May and June 2024. The participants were recruited online via various social media platforms.

Eligible were primary family caregivers aged 18 years or older, residing in Lebanon, and providing unpaid care for community-dwelling older adults (65 years or older). Primary family caregivers were defined as spouses, children, or close relatives with basic literacy skills. Individuals not meeting these criteria were categorized as controls.

**Data collection.** Demographic information and additional attributes were obtained from caregivers through an online survey created on Google Forms and distributed on social media via the snowball technique. The two-part questionnaire was available in Arabic, Lebanon's native language.

**Sample size calculation.** Statistical guidelines recommend a respondent-to-item ratio of 10:1 for exploratory factor analysis (EFA) [50] and a minimum of 200 participants for confirmatory factor analysis (CFA) [51]. In the present study, the recommended ratio was applied, resulting in a sample size of 310 participants who completed an 11-item questionnaire for the factor analysis.

**Instruments.** Caregivers were requested to fill out a questionnaire in two sections. The first section collected caregivers' sociodemographic data, including age, gender, marital status, educational level, monthly household income, work status, and relationship with the older adult (spouse, child, daughter-in-law, son-in-law, grandchild, or other relative).

The second part of the study focused on collecting data on the demographic and other characteristics of care recipients. Caregivers were requested to provide the following information: demographic details (age, gender, and educational level) and clinical data, including medical diagnoses and a list of chronic diseases such as dementia and stroke. Caregivers also reported on care recipients' basic functional abilities and frailties via two validated scales in addition to instrumental activities in daily living via the ADF-CS. Permissions for the use of all these scales were secured for the study.

A. **The Katz Index of Activities of Daily Living (ADL)** [52]. This 6-item tool was used to assess performance in basic activities of daily living required for independence in self-care (e.g., hygiene, dressing, eating, and continence). The validity of the Arabic version of the ADL scale has been demonstrated among Lebanese older adults living in nursing homes [53]. Responses to this scale were rated as 0, 0.5, or 1, with 0.5 denoting partial independence. Higher scores indicate greater functional independence in performing basic self-care activities.

B. **The Clinical Frailty Scale (CFS).** This simple and accessible tool enables a quick and straightforward assessment of frailty in older individuals [54]. It consists of an ordinal 9-point scale represented visually through a chart to aid in categorizing frailty levels. The scale ranges from very fit (CFS = 1) to very severely frail (CFS = 8) and terminally ill (CFS = 9) on the basis of descriptions and pictorial representations of activity and functional status. Frailty is typically defined as a CFS score greater than 4 [55]. Higher scores indicate greater frailty and associated risks [54].

**Data management and analysis plan.** The statistical software SPSS version 27.0 was used for data entry and analysis. Descriptive statistics were reported as medians and interquartile ranges (IQRs) for continuous variables and as frequencies (n) with percentages (%) for categorical variables. Cronbach's alpha was used to evaluate the scale's internal consistency, with coefficients above 0.7 considered acceptable. On the basis of previous investigations [56], it was assumed that IADL performance is positively associated with frailty and ADL disability.

Exploratory factor analysis (EFA) and confirmatory factor analysis (CFA) were carried out to assess factorial validity. These analyses were conducted for the ADF-CS using random split-half samples. EFA was performed on the first random-half subsample to explore the factor structure via principal component analysis with varimax rotation. Sampling adequacy was determined via the Kaiser–Meyer–Olkin (KMO) measure and Bartlett's test of sphericity. The number of factors retained was based on eigen values greater than one and visual inspection of the scree plot. Items with low communalities (less than 0.3) or high cross-loadings were removed. CFA was performed through structural equation modeling, with maximum likelihood estimation used to examine the fit of the data to the ADF-CS factor structure. Model fit was considered adequate when the relative chi-square index ($\chi^2$/ df) was ≤ 3.0, the comparative fit index (CFI) was > 0.90, the goodness-of-fit index (GFI) was > 0.90, and the root mean square error of approximation (RMSEA) was < 0.08 [57].

There was a suggestion to exclude men's food preparation, housekeeping, and laundry items, with reasons provided, while advocating for the complete scale's use for women [15,58]. However, the structural validity of the Lawton IADL scale has not been verified. Therefore, both EFA and CFA of the ADF-CS were conducted separately for females and males, including all the items.

The known group validity of the ADF-CS was assessed by comparing its scores across different age groups and the presence of specific chronic diseases (dementia and stroke) as reported by the caregiver. It was hypothesized that older age groups would have lower ADF-CS scores than younger age groups. Additionally, older adults with both dementia and stroke were expected to have lower ADF-CS scores (greater disability) than those with only one of these conditions. The effect size for statistically significant differences was determined via Cohen's D, where 0.2 represents a small effect size, 0.5 represents a medium effect size, and 0.8 or above represents a large effect size [59].

Convergent validity was tested via the Spearman correlation coefficient. The significance level for all the statistical tests was set at a P-value < 0.05 (two-sided).

## Phase IV: Assessing the agreement of responses between self-reported and informant-reported ADF tasks

**Study design and participants.** A cross-sectional study was conducted in Lebanon in July 2024 to assess the agreement in responses to the ADF-CS between caregivers and older adults. Face–to–face interviews were carried out by the principal investigator with caregivers and their respective care recipients. Care recipients (older adults) were eligible for inclusion if they (1) lived in the community, (2) were 65 years or older, or (3) had a family member caregiver present during the interview. The exclusion criteria were as follows: (1) age under 65 years; (2) a diagnosis of cognitive impairment, schizophrenia, or organic hallucinosis; (3) acute delirium; (4) aphasia or significant speech and language impairments as observed during the interview; (5) severe hearing impairments not compensated by a hearing aid; and (6) severe behavioral symptoms.

Caregivers were subjected to the same eligibility criteria as those in the third phase of the study, with the additional requirement of providing care to older adults who met the specified criteria. Caregivers who refused to provide accurate and reliable information, as outlined in the consent form, were excluded.

**Sample size calculation.** The required sample size was determined via G*Power software. A correlation coefficient was computed to evaluate reliability, assuming a moderate to large effect size (r = 0.4). Considering a 5% alpha error and a 20% beta error, the minimum sample size needed was 44 pairs of participants.

**Instruments.** Demographic characteristics, including age, gender, educational level, and caregiver-older adult kinship, as well as ADF-CS responses, were collected separately for both caregivers and care recipients.

**Data collection.** Participants were recruited via snowball sampling to assess the concordance of responses to the ADF-CS between caregivers and older adults. The principal investigator initially recruited caregivers and their care recipients from her personal network. Through referrals, each participant helped identify additional caregiver–care recipient pairs, expanding the sample. Both care recipients and their respective caregivers provided demographic information and completed the ADF-CS. The questionnaire was administered in Arabic to both groups via paper-based methods. Unique identification numbers (e.g., 1–44) were assigned to each participant instead of personal details to ensure anonymity. All the data were securely stored, with limited access granted to authorized research staff. Data summaries were shared during analysis and reporting without disclosing individual identities.

**Data management and analysis plan.** The intraclass correlation coefficient (ICC; average measure) was calculated to assess the agreement of responses on the ADF-CS across 44 pairs of participants. ICC values can range between 0 and 1, with scores between 0.40 and 0.59 indicating fair agreement, 0.60 to 0.74 indicating good agreement, and 0.75 to 1 indicating excellent agreement [60]. Additionally, Cohen's kappa correlation coefficient was computed for this study phase. Kappa values between 0.2 and 0.4 reflect a fair agreement, 0.41 to 0.6 reflect a moderate agreement, 0.61 to 0.8 reflect a substantial agreement, and 0.81 to 1 indicate an almost perfect agreement [61]. Statistical significance was set at P-value < 0.05 (two-sided).

## Results

### Phases I and II: Development, translation, and cross-cultural adaptation of the ADF-CS

Following a rigorous methodology, the panel of nine experts refined the ADF-CS to 12 items, which were then translated and cross-culturally adapted to produce a final Arabic version of the scale. A pilot test was carried out with 15 participants to assess the clarity and appropriateness of the Arabic ADF-CS. The participants completed the questionnaire in five minutes and did not express any issues regarding item clarity or comprehensibility.

### Phase III: Psychometric evaluation of the ADF-CS

**Study participant characteristics.** A total of 544, with a median age of 40 years (IQR = 15), participated in the study. Among them, 84% were female, 66.9% were married, and over one-third held a university degree (36.9%). Additionally, 243 caregivers (44.7%) were unemployed, and 241 (44.3%) had a monthly household income of less than 250 USD. With respect to kinship with the care recipient, 61.9% were daughters or sons, 17.8% were daughters-in-law or sons-in-law, and 26.8% were spouses (Table 1).

Care recipients were predominantly female (70.4%) and were reported by caregivers as either illiterate (44.5%) or having only an elementary education (29.4%). The median age of the care recipients was 75.5 years (IQR = 15). Among those, 24.6% reported having dementia, and 12.3% had experienced a stroke (Table 2).

**Reliability of the Arabic ADF-CS.** The internal consistency of the ADF-CS was assessed using Cronbach's alpha. The overall Cronbach's alpha for the ADF-CS was 0.901, with the first and second factors showing alphas of 0.829 and 0.871, respectively. This high internal consistency indicates that both factors, as well as the overall ADF-CS, effectively measure the same underlying construct. The correlation coefficients between individual items and the total score ranged from 0.515 to 0.710, all exceeding 0.4. Removing any single item did not significantly affect Cronbach's alpha, which ranged from 0.888 to 0.898 (Table 3).

**Validity of the ADF-CS.** *Convergent validity:* A strong positive correlation was found between the total ADF-CS score and the ADL score, with a correlation coefficient of 0.815. Additionally, a significant difference in ADF-CS means was observed across the three levels of frailty ($p < 0.0001$), providing evidence for convergent validity (Table 4).

*Structural validity:* **Exploratory factor analysis**. The validity of the ADF-CS was assessed via exploratory factor analysis with 12 items. The KMO measure indicated sampling adequacy at 0.887, and the χ2 test ($\chi^2 = 1677.41$, df = 55; $p < 0.0001$) confirmed the appropriateness of the factor analysis. All the items were successfully extracted except for item 7 (ability to use electronic devices such as computers and tablets), which displayed low communality and was subsequently removed. The EFA was then repeated with the remaining 11 items, yielding a two-factor solution with an eigenvalue exceeding 1, explaining 63.13% of the variance. Parallel analysis supported the retention of these two factors.

After extensive discussions between the expert panel and the research team, factor 1 was labeled "ordinary and usual daily tasks" and included five items: food preparation, shopping, housekeeping, traveling alone, and operating electrical household appliances. Factor 2, termed "high dexterity and cognitive ability-demanding tasks," reflected tasks requiring higher levels of dexterity and cognitive ability and comprised six items: using the landline telephone, using a mobile phone, using television, managing medication, handling finances, and going to places beyond walking distance.

The gender-specific analysis yielded consistent results, with an equal number of factors and low communality for item 7. The factor loadings for the 11 items ranged from 0.506 to 0.904 for males and from 0.516 to 0.905 for females. The communalities varied from 0.414 to 0.837 for males and from 0.365 to 0.853 for females. The factors accounted for 62.05% of the variance for males and 63.09% for females. Consequently, findings for the overall sample were presented (Table 5).

**Confirmatory factor analysis.** A confirmatory factor analysis was conducted using the two-factor structure identified in the EFA. The initial model did not meet the required fit criteria, necessitating adjustments by creating covariance for error terms on the basis of modification indices exceeding 30. Introducing a path between ADF-CS1 and ADF-CS2 improved

**Table 1. Sociodemographic characteristics of the caregivers (N = 544).**

| Variables | n | % |
|---|---|---|
| **Age (years) Median (IQR)** | 40 (15) | |
| **Gender** | | |
| Male | 87 | 16 |
| Female | 457 | 84 |
| **Marital Status** | | |
| Married | 364 | 66.9 |
| Single | 111 | 20.4 |
| Divorced | 53 | 9.7 |
| Widowed | 16 | 2.9 |
| **Relationship with older adults** | | |
| Son/daughter | 337 | 61.9 |
| Son-in-law/daughter-in-law | 97 | 17.8 |
| Spouse | 12 | 2.2 |
| Siblings | 8 | 1.5 |
| Others | 90 | 16.6 |
| **Educational Level** | | |
| Elementary or lower | 40 | 7.4 |
| Intermediate | 141 | 25.9 |
| Secondary | 120 | 22.1 |
| University | 201 | 36.9 |
| Postgraduate | 42 | 7.7 |
| **Working Status** | | |
| Work | 301 | 55.3 |
| Do not work | 243 | 44.7 |
| **Household Monthly Income** | | |
| <250 USD | 241 | 44.3 |
| 250-500 USD | 165 | 30.3 |
| 500-1000 USD | 87 | 16 |
| 1000-2000 USD | 38 | 7 |
| >2000 USD | 13 | 2.4 |

IQR: Interquartile range

the model fit. The final model demonstrated favorable fit indices: RMSEA = 0.079 (< 0.10) and maximum likelihood $\chi^2 = 110.379$ with df = 42, resulting in a $\chi^2/df$ ratio of 2.63 (< 3). The Tucker–Lewisindex (TLI) and comparative fit index (CFI) values were 0.939 and 0.953, respectively (> 0.9). The Jöreskog goodness-of-fit index (GFI) was 0.928, and the adjusted GFI was 0.887. All standardized factor loadings were statistically significant (p < 0.001), ranging from 0.60 to 0.80 (**Fig 1**).

When stratified by sex, the CFA fit indices remained favorable. The RMSEA was 0.076 for males and 0.067 for females. The GFI and CFI values were 0.947 and 0.967 for females and 0.924 and 0.960 for males. All factor loadings were statistically significant (p < 0.001), ranging from 0.53 to 0.90 for males and 0.53 to 0.84 for females.

**Known-group validity.** Known-group validity was supported by statistically significant differences in ADF-CS means across different age groups and was based on the reported presence of specific chronic diseases (**Table 6**). An ordered change in ADF-CS scores was observed after stratification, with older individuals showing lower physical function than

**Table 2. Caregiver-reported baseline and clinical characteristics of care recipients (N = 544).**

| Variables | All (N = 544) | |
|---|---|---|
| | n | % |
| **Gender** | | |
| Male | 161 | 29.6 |
| Female | 383 | 70.4 |
| **Educational Level** | | |
| Illiterate | 242 | 44.5 |
| Elementary | 160 | 29.4 |
| Intermediate | 80 | 14.7 |
| Secondary | 27 | 5 |
| University | 35 | 6.4 |
| **Dementia** | | |
| Absent | 410 | 75.4 |
| Present | 134 | 24.6 |
| **Stroke** | | |
| Absent | 477 | 87.7 |
| Present | 67 | 12.3 |
| | Median | IQR |
| **Age (years)** | 75.5 | 15 |
| **ADF-CS** | 18 | 8 |
| **ADL** | 4 | 3.5 |

n: frequency; %: percentage; IADL: instrumental activities of daily living; IQR: interquartile range

**Table 3. Internal Consistency of the Arabic ADF-CS (N = 544).**

| | Scale Mean if Item Deleted | Scale Variance if Item Deleted | Corrected Item-Total Correlation | Cronbach's Alpha if Item Deleted |
|---|---|---|---|---|
| **ADF-CS1** | 16.79 | 24.66 | 0.664 | 0.890 |
| **ADF-CS2** | 16.86 | 24.83 | 0.648 | 0.891 |
| **ADF-CS3** | 17.04 | 26.11 | 0.592 | 0.894 |
| **ADF-CS4** | 17.31 | 25.93 | 0.682 | 0.890 |
| **ADF-CS5** | 17.25 | 26.30 | 0.612 | 0.894 |
| **ADF-CS6** | 16.78 | 23.79 | 0.683 | 0.890 |
| **ADF-CS8** | 17.30 | 25.71 | 0.640 | 0.892 |
| **ADF-CS9** | 17.26 | 26.32 | 0.590 | 0.895 |
| **ADF-CS10** | 16.65 | 23.95 | 0.710 | 0.888 |
| **ADF-CS11** | 17.11 | 25.03 | 0.697 | 0.889 |
| **ADF-CS12** | 16.82 | 26.48 | 0.515 | 0.898 |

ADF-CS: Autonomous in Daily Functioning-Contemporary Scale

younger individuals. Additionally, care recipients with dementia had worse outcomes than those without dementia, and those with both dementia and stroke exhibited poorer functioning than those with either dementia or stroke alone. The effect size ranged from 0.55 to 1.20, with the greatest effect sizes observed in the age group comparison of ≤ 75 versus > 85 years and the presence versus absence of dementia.

**Table 4. Convergent validity of the ADF-CS total score.**

| Variables | N | Mean (SD) | r | P-value |
|---|---|---|---|---|
| **ADL** | | | 0.815 | **<0.0001** |
| **Frailty** | | | | **<0.0001** |
| Strong | 90 | 24.4 (5.43) | | |
| Weak | 323 | 18.82 (4.74) | | |
| Dependent | 131 | 14.56 (3.36) | | |

N: frequency; SD: standard deviation; r: correlation coefficient; a p-value of < 0.05 was considered significant.

**Table 5. Factor Structure of the ADF-CS.**

| Items | Factor 1 | Factor 2 | Communality |
|---|---|---|---|
| Ability to prepare food (**ADF-CS8**) | 0.788 | | 0.664 |
| Shopping ability (**ADF-CS4**) | 0.734 | | 0.629 |
| Housekeeping ability (**ADF-CS5**) | 0.721 | | 0.595 |
| Ability to travel abroad alone (**ADF-CS9**) | 0.719 | | 0.555 |
| Ability to use electrical household appliances (**ADF-CS12**) | 0.718 | | 0.521 |
| Ability to use the landline telephone (**ADF-CS1**) | | 0.917 | 0.855 |
| Ability to use a mobile phone (**ADF-CS2**) | | 0.883 | 0.788 |
| Ability to use television (**ADF-CS10**) | | 0.699 | 0.679 |
| Ability to manage medication (**ADF-CS6**) | | 0.620 | 0.587 |
| Ability to handle finances (**ADF-CS11**) | | 0.545 | 0.592 |
| Ability to go to places beyond walking distance (**ADF-CS3**) | | 0.493 | 0.480 |
| **Eigenvalue** | 5.56 | 1.38 | |
| **Percentage of variance** | 50.59 | 12.54 | |

### Phase IV: Assessing the agreement of responses between self-reported and informant-reported ADF tasks

A sample of 44 caregiver/care recipient pairs participated in this study phase. The intraclass and Cohen's kappa correlation coefficient assessments of the ADF-CS demonstrated excellent agreement in responses between caregivers and their respective care recipients (**Table 7**).

## Discussion

The limitations of current IADL assessment tools and the lack of a culturally relevant version for Arabic-speaking populations prompted this research. The present study aimed to develop, cross-culturally adapt, and validate an updated version of the traditional Lawton IADL scale, the ADF-CS. The newly developed ADF-CS demonstrated good psychometric properties and is thus effective for assessing functional autonomy in Arabic-speaking community-dwelling older adults.

Owing to the innovative nature of the developed scale, direct comparisons with other studies posed challenges. Although the Lawton IADL scale is widely regarded as suitable for use by either care recipients or knowledgeable family members or caregivers [15], most validation studies focus solely on self-reports from older adults [19–21,23,24,27,29,30], with limited research comparing self-reported abilities with those reported by informants or caregivers of older adults. Given that some older adults may be unable to complete the scale independently due to physical or mental impairments, it was deemed essential to develop and validate an updated informant-based version of the Lawton IADL scale, allowing caregivers to complete the scale reliably on behalf of care recipients.

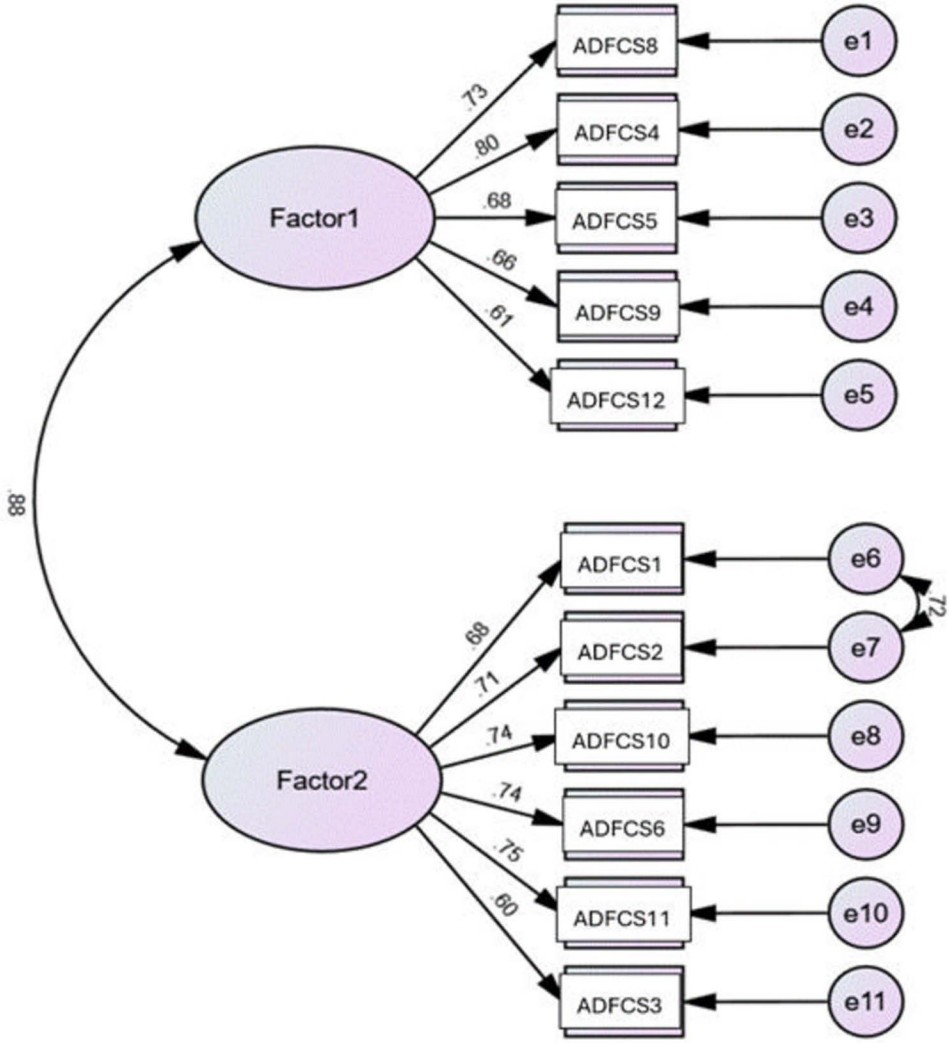

**Fig 1. Standardized factor loadings of the two-factor model of the Arabic ADF-CS.** *Factor 1: ordinary and usual daily tasks; Factor 2: high dexterity and cognitive ability demanding tasks; errors 1 to 11 are unobserved variables.*

The findings revealed that the internal consistency of the overall ADF-CS and its two factors was good to excellent, suggesting that the items within each dimension consistently measured the intended construct. The Cronbach's alpha for the overall ADF-CS is comparable to those reported for the AIADL scale [41], as well as the self-reported [19,29] and informant-reported measures of the Lawton IADL scale [22,27]. With respect to face and content validity, the respondents reported no difficulties understanding the scale, leading to no further modifications. Convergent validity was supported by the strong positive correlation between the total ADF-CS score and the ADL disability score (r = 0.815). This correlation is stronger than those observed in the validation of the Spanish (r < 0.73) [19] and Sinhala (r = 0.61) [29] self-reported versions of the Lawton IADL scale. A statistically significant correlation was also found between the ADF-CS total score and frailty. As expected, frailty in older individuals was associated with reduced autonomous functioning, as previously reported [62]. This result confirms the convergent validity of the Arabic version of the ADF-CS.

**Table 6. Mean ADF-CS scores by age group and presence of chronic conditions in older adults.**

| Variable | n | Mean (SD) | P-value | Effect size |
|---|---|---|---|---|
| **N = 538** | | | | |
| **Age** | | | **<0.0001** | |
| (≤75)[a] | 269 | 20.75 (5.58) | (<0.0001)[a,c] | (1.20)[a,c] |
| (76–85)[b] | 169 | 17.82 (4.84) | (<0.0001)[b,a] | (0.55)[b,a] |
| (>85)[c] | 100 | 14.64 (3.42) | (<0.0001)[c,b] | (0.72)[c,b] |
| **N = 544** | | | | |
| **Dementia** | | | **<0.0001** | 1.16 |
| Absent | 410 | 20.12 (5.22) | | |
| Present | 134 | 14.40 (3.93) | | |
| **N = 170** | | | | |
| **Stroke and Dementia** | | | 0.003 | 0.63 |
| Combined stroke and dementia | 31 | 12.90 (1.74) | | |
| Either stroke or dementia | 139 | 15.79 (4.98) | | |

N, n: frequency; SD: standard deviation; superscript letters a, b, and c denote differences between the corresponding age group categories; and a P-value less than 0.05 was considered significant.

**Table 7. Intraclass and Cohen's kappa correlation coefficients for ADF-CS items.**

| Variables | Intraclass correlation coefficient (95%CI) | P-value | Kappa correlation coefficient | P-value |
|---|---|---|---|---|
| Ability to use the landline telephone (**ADF-CS1**) | 0.994 (0.989, 0.997) | **<0.0001** | 0.945 | **<0.0001** |
| Ability to use a mobile phone (**ADF-CS2**) | 1 | | 1 | **<0.0001** |
| Navigating beyond walking distance (**ADF-CS3**) | 1 | | 1 | **<0.0001** |
| Shopping ability (**ADF-CS4**) | 1 | | 1 | **<0.0001** |
| Housekeeping ability (**ADF-CS5** | 1 | | 1 | **<0.0001** |
| Medication Management (**ADF-CS6**) | 1 | | 1 | **<0.0001** |
| Ability to prepare food (**ADF-CS8**) | 1 | | 1 | **<0.0001** |
| Solo travel abroad (**ADF-CS9**) | 0.974 (0.953, 0.986) | **<0.0001** | 0.949 | **<0.0001** |
| Ability to use television (**ADF-CS10**) | 1 | | 1 | **<0.0001** |
| Ability to handle finances (**ADF-CS11**) | 1 | | 1 | **<0.0001** |
| Ability to use electrical household appliances (**ADF-CS12**) | 0.982 (0.967, 0.990) | **<0.0001** | 0.929 | **<0.0001** |
| **Total ADF-CS score** | 0.999 (0.998, 1.000) | **<0.0001** | | |

In the exploratory factor analysis, all the items were successfully extracted except for the ADF-CS7, which showed low communality and was subsequently removed. The revised analysis identified two psychometrically robust subdomains, "ordinary and usual daily tasks" and "high dexterity and cognitive ability-demanding tasks," explaining 63.13% of the scale's total variance. The findings of the factorial analysis were consistent across both sexes, aligning with the results of other studies on Lawton IADL validation [19,29]. Previous research has often revealed unidimensionality in the Lawton IADL scale's structure [19,23], although two studies identified a two-factor structure [28,63]. The first categorized the scale into physical (preparing meals, doing laundry, housekeeping, going to places outside the house, and shopping) and cognitive (managing finances, taking medications, and using the phone) IADL domains, whereas the second revealed two factors: "home living and management" and "community living."

The CFA of the ADF-CS demonstrated that all the items within each component fit well with their intended constructs, maintaining the integrity of the two-factor structure. Removing any item was unnecessary, and the 11-item structure

remained consistent across both sexes, supporting its applicability for evaluating functional abilities in older individuals of both genders. This consistency aligns with previous validations of the Lawton IADL scale [19,29], which indicated that the full scale can be used for both males and females.

Known-group validity testing of the ADF-CS revealed statistically significant variations in mean scores across age groups, with a gradual decline in scores as age increased. This pattern is consistent with findings from the Lawton IADL scale [19,63], confirming that the ADF-CS effectively distinguishes functional differences between older adults of different ages. The ADF-CS scores were also lower in participants with dementia than in those without dementia, indicating that the ADF-CS could also differentiate between people on the basis of their cognitive status. This finding echoes earlier results showing that the Lawton IADL scale discriminates well between older adults with and without dementia [64]. Furthermore, participants with both dementia and stroke exhibited lower functionality than those with either of these conditions alone, which is consistent with previous findings that the combination of stroke and dementia reduces functional ability compared with either condition alone [65,66].

Moreover, excellent intraclass and Cohen's kappa correlation coefficients were observed, indicating a near-perfect response agreement between self-reported and informant-reported responses. This robust concordance further validates the ADF-CS and supports its use by family caregivers on behalf of older adults [12,67].

The present study has several limitations. First, the test-retest reliability of the ADF-CS was not assessed. Second, selection bias is possible due to the use of snowballs and convenience sampling methods. Third, the use of self-reports or surrogate reporting methods of administration rather than a demonstration of the functional task could result in either an overestimation or underestimation of abilities. Additionally, the tool may not be sensitive to minor, incremental changes in function. Future research is needed to explore the instrument's ability to detect changes resulting from clinical intervention over time.

Moreover, the innovative nature of the developed scale impedes result comparability, thus limiting its generalizability. Another limitation is that stroke and dementia were assessed by caregivers' declared questions rather than by clinical evaluations or validated tools to ensure accurate and reliable diagnoses. Finally, future research should also examine the psychometric properties of our scale in other Arab countries to ensure that the validity and reliability of the scale are replicable and well-established.

Despite these limitations, our study provides preliminary evidence indicating that the Arabic version of the ADF-CS has good psychometric properties and can be used to assess functional independence in Arabic-speaking community-dwelling older individuals living in contemporary settings. Evaluating the inability to perform IADLs in elderly individuals is crucial for determining the necessary level of assistance and serves as an essential measure for programs and services aimed at caring for older individuals [41]. Moreover, assessing functional abilities and working with practitioners on a care plan to overcome barriers that hinder the performance of older people is imperative for aging in place, as older adults opt to remain in their homes rather than being placed in institutions [68].

## Conclusion

The Arabic version of the Autonomy in Daily Functioning-Contemporary Scale is a reliable and valid informant-reported measure for assessing instrumental activities of daily living in community-dwelling older adults. This validated version can assist rehabilitation and occupational therapy professionals in evaluating older adults' ability to perform daily activities in a contemporary community.

## Supporting information

**S1 File.  Flowchart of the study phases.**
(DOCX)

**S2 File.  ADF-C Scale English-Arabic version.**
(DOCX)

**S3 File. Details of the five-step methodology utilized to translate the scales.**
(DOCX)

**S4 File. Study dataset.**
(SAV)

## Acknowledgments

The authors would like to thank all the caregivers and care recipients who participated in this study.

## Author contributions

**Conceptualization:** Zainab Barakat, Sarah Khatib, Carmela BouMalham, Chadia Haddad, Marwan Akel, Samar Rachidi, Pascale Salameh.

**Data curation:** Zainab Barakat.

**Formal analysis:** Zainab Barakat.

**Investigation:** Zainab Barakat.

**Methodology:** Zainab Barakat, Sarah Khatib, Carmela BouMalham, Chadia Haddad, Pascale Salameh.

**Project administration:** Zainab Barakat.

**Supervision:** Linda Abou Abbas, Marc Barakat, Samar Rachidi, Pascale Salameh.

**Validation:** Zainab Barakat.

**Visualization:** Zainab Barakat.

**Writing – original draft:** Zainab Barakat.

**Writing – review & editing:** Hala Sacre, Aline Hajj, Rony M Zeenny, Linda Abou Abbas, Marc Barakat, Samar Rachidi.

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
