## [Decision Letter · Decision Letter 0]

12 Mar 2025

PONE-D-24-52323A Contemporary Tool for Assessing Instrumental Activities of Daily Living: Validation of a Caregiver-Reported Scale for Non-Institutionalized Older AdultsPLOS ONE

Dear Dr. Barakat,

Thank you for submitting your manuscript to PLOS ONE. After careful consideration, we feel that it has merit but does not fully meet PLOS ONE’s publication criteria as it currently stands. Therefore, we invite you to submit a revised version of the manuscript that addresses the points raised during the review process.

We look forward to receiving your revised manuscript.

Kind regards,

Maria José Nogueira, Ph.D.

Academic Editor

PLOS ONE

**Journal Requirements:**

Please ensure that your manuscript meets PLOS ONE's style requirements, including those for file naming. The PLOS ONE style templates can be found at https://journals.plos.org/plosone/s/file?id=wjVg/PLOSOne_formatting_sample_main_body.pdf and https://journals.plos.org/plosone/s/file?id=ba62/PLOSOne_formatting_sample_title_authors_affiliations.pdf 2. In the online submission form, you indicated that Data are available from the corresponding authors upon reasonable request.  All PLOS journals now require all data underlying the findings described in their manuscript to be freely available to other researchers, either a. In a public repository, b. Within the manuscript itself, or c. Uploaded as supplementary information.This policy applies to all data except where public deposition would breach compliance with the protocol approved by your research ethics board. If your data cannot be made publicly available for ethical or legal reasons (e.g., public availability would compromise patient privacy), please explain your reasons on resubmission and your exemption request will be escalated for approval.

**Additional Editor Comments:**

Dear editor,

The article is interesting, but needs a minor revision.

Best Regards

Reviewers' comments:

Reviewer's Responses to Questions

**Comments to the Author**

1. Is the manuscript technically sound, and do the data support the conclusions?

Reviewer #1: Yes

2. Has the statistical analysis been performed appropriately and rigorously?

Reviewer #1: Yes

3. Have the authors made all data underlying the findings in their manuscript fully available?

Reviewer #1: Yes

4. Is the manuscript presented in an intelligible fashion and written in standard English?

Reviewer #1: No

5. Review Comments to the Author

**Reviewer #1: ** Dear Authors,

I read your work entitled “A Contemporary Tool for Assessing Instrumental Activities of Daily Living: Validation of a Caregiver-Reported Scale for Non-Institutionalized Older Adults.” and here I enclose my recommendations to you:

1. There is a need for editing many English language errors. Please, have a more thorough “look” in the text by a native speaker of English or an editing office.

2. The “Introduction” is poor and the rational of this study is not clear. The Authors have to address a lot in this work since they targeted to specific population. The translated versions of Lawton’s IADL are not all reported, is suggest the Authors to include them see a quick search I have performed for consultation (https://scholar.google.com/scholar?hl=el&as_sdt=0%2C5&q=lawton%27s+iadl+scale+validation&btnG=
https://scholar.google.com/scholar?hl=el&as_sdt=0%2C5&q=lawton%27s+iadl+scale+validation+greek&btnG=).

3. The “Methods and the Results” are readers friendly and I congratulate the Authors for that.

4. The discussion also must be oriented to the population included in this study. If the Authors will updated their Introduction as suggested above they can enrich their discussion with more studies and strengthen more this sectiopn

Thank you!

6. PLOS authors have the option to publish the peer review history of their article (what does this mean? ). If published, this will include your full peer review and any attached files.

**Do you want your identity to be public for this peer review?** For information about this choice, including consent withdrawal, please see our Privacy Policy .

Reviewer #1: No

---

## [Author Response · Author response to Decision Letter 1]

21 Mar 2025

March 20, 2025

Dear Dr Maria José Nogueira,

We are pleased to resubmit the revised version of Manuscript entitled: “A contemporary tool for assessing instrumental activities of daily living: validation of a caregiver-reported scale for non-institutionalized older adults.”

Ref: Submission ID 36cd00b93930ffc3

We would like to thank the editor and the reviewer for their very helpful comments. We have carefully addressed all the comments. We hope our revision has improved the paper to a level of their satisfaction. Our responses are in italics and are prefaced by “Author response.” Corresponding changes are highlighted in the revised file. Please find attached our revised paper, and below is a summary of how we responded to the comments. My coauthor and I appreciate the opportunity to resubmit and are excited about the possibility of publication.

Editor

Thank you for your time and editorial guidance!

Author response: Thank you for your comment! I have reviewed the PLOS ONE style requirements, including those for file naming, and can confirm that the manuscript complies with these guidelines. Please let me know if any additional revisions are needed.

2. In the online submission form, you indicated that Data are available from the corresponding authors upon reasonable request. All PLOS journals now require all data underlying the findings described in their manuscript to be freely available to other researchers, either a. In a public repository, b. Within the manuscript itself, or c. Uploaded as supplementary information.

Author’s response: Uploaded as supplementary information.

Author’s response: Reviewed and verified. No retracted papers have been cited.

Reviewer

I read your work entitled “A Contemporary Tool for Assessing Instrumental Activities of Daily Living: Validation of a Caregiver-Reported Scale for Non-Institutionalized Older Adults.”

Thank you for taking time to review the article and for providing such valuable input!

Here I enclose my recommendations to you:

1. There is a need for editing many English language errors. Please, have a more thorough “look” in the text by a native speaker of English or an editing office.

Author response: Thank you for your comment! As requested, the English language errors have been corrected by a native English speaker as requested.

2. The “Introduction” is poor and the rational of this study is not clear. The Authors have to address a lot in this work since they targeted to specific population. The translated versions of Lawton’s IADL are not all reported, is suggest the Authors to include them see a quick search I have performed for consultation (https://scholar.google.com/scholar?hl=el&as_sdt=0%2C5&q=lawton%27s+iadl+scale+validation&btnG=
https://scholar.google.com/scholar?hl=el&as_sdt=0%2C5&q=lawton%27s+iadl+scale+validation+greek&btnG=).

Author response: Thank you for pointing out these issues! I have revised the introduction to clearly express the rationale of the study. Additionally, I have reviewed and updated the references of the translated versions of the IADL Lawton as requested. Please refer to the revised manuscript for the updates. I would greatly appreciate any further suggestions or if you require additional modifications.

3. The “Methods and the Results” are readers friendly and I congratulate the Authors for that.

Author response: Thank you for your valuable feedback!

4. The discussion also must be oriented to the population included in this study. If the Authors will updated their Introduction as suggested above they can enrich their discussion with more studies and strengthen more this section.

Author response: Thank you for your comment! Our study developed a new scale, which makes direct comparisons with other studies challenging. However, we have made careful efforts to include relevant comparisons where possible. As requested, the discussion has been revised and encriched with additional studies. Please refer to the revised manuscript for the updates. I would greatly appreciate any further suggestions or if you require any additional modifications.

---

## [Editor Report · Decision Letter 1]

24 Mar 2025

A contemporary tool for assessing instrumental activities of daily living: validation of a caregiver-reported scale for non-institutionalized older adults

PONE-D-24-52323R1

Dear Dr. Zainab Barakat

We’re pleased to inform you that your manuscript has been judged scientifically suitable for publication and will be formally accepted for publication once it meets all outstanding technical requirements.

Kind regards,

Maria José Nogueira, Ph.D.

Academic Editor

PLOS ONE

Additional Editor Comments (optional):

Dear Author

Revisions increased the quality of the article.

The article is now able to be considered for publication.
---

## [Editor Report · Acceptance letter]

PONE-D-24-52323R1

PLOS ONE

Dear Dr. Barakat,

I'm pleased to inform you that your manuscript has been deemed suitable for publication in PLOS ONE. Congratulations! Your manuscript is now being handed over to our production team.

Kind regards,

on behalf of

Professor Maria José Nogueira

Academic Editor

PLOS ONE